# MW-PPG Sensor: An on-Chip Spectrometer Approach

**DOI:** 10.3390/s19173698

**Published:** 2019-08-26

**Authors:** Cheng-Chun Chang, Chien-Ta Wu, Byung Il Choi, Tong-Jing Fang

**Affiliations:** 1Department of Electrical Engineering, National Taipei University of Technology, Taipei 10608, Taiwan; 2nanoLambda Co. Ltd., Daejeon 34141, Korea; 3Department of Physiology and Biophysics, Graduate Institute of Physiology National Defense Medical Center, Taipei 11490, Taiwan

**Keywords:** multi-wavelengths, MW-PPG, on-chip spectrometers

## Abstract

Multi-wavelength photoplethysmography (MW-PPG) sensing technology has been known to be superior to signal-wavelength photoplethysmography (SW-PPG) sensing technology. However, limited by the availability of sensing detectors, many prior studies can only use conventional bulky and pricy spectrometers as the detectors, and hence cannot bring the MW-PPG technology to daily-life applications. In this study we developed a chip-scale MW-PPG sensor using innovative on-chip spectrometers, aimed at wearable applications. Also in this paper we present signal processing methods for robustly extracting the PPG signals, in which an increase of up to 50% in the signal-to-noise ratio (S/N) was observed. Example measurements of saturation of peripheral blood oxygen (SpO_2_) and blood pressure were conducted.

## 1. Introduction

Photoplethysmography (PPG) has been a commonly-used optical sensing method that collects light reflected or transmitted through skin so as to noninvasively monitor the pulsation of blood flow in subcutaneous blood vessels. Since blood flow pulsations can reflect the operating conditions of the circulatory and respiratory systems of the human body, PPG signals can be used as indicators for many diseases, such as endothelial dysfunction, sympathetic neuropathy, cardiac arrhythmia, vasospasm, microcirculation, autonomic neuropathy, orthostatic hypotension, migraine, and peripheral artery disease [1,2,3,4]. Due to the simple measurement structure, PPG sensing technology has been widely used in wearable devices to achieve heart rate detection in recent years [5,6,7,8,9,10]. In 2018, the penetration rate of PPG sensing technology in wearable devices reached 98% and it is expected to reach 100% by 2020 [11]. It is expected that the global net profit of wearable devices will reach 52.5 billion U.S. dollars in 2024 [12].

Currently, PPG sensing devices using single-wavelength (SW) light sources have been the main stream on the market. Many studies have focused on reducing the effects of motion artifacts, mostly by using accelerometers to build up compensation signals, hence improving the signal-to-noise ratio (S/N) of the PPG signals [13,14,15]. However, heart rate measurement error can be still up to 10% using the single-wavelength PPG (SW-PPG) sensing technology. Besides, the SW-PPG sensing technology may also suffer from many other factors during measurement, such as skin color, skin surface temperature and sensor contact pressure, resulting in a poor quality of PPG signals.

Accordingly, multi-wavelength PPG (MW-PPG) sensing technology has gradually attracted the attention of many scholars in recent years, and has been considered a robust PPG signal measurement method [16,17]. In earlier studies [18,19], it has been noted that PPG sensing light sources at different wavelengths are recommended for the subjects with different skin colors. In applying multi-wavelength photoplethysmography (MW-PPG) sensing technology, the most suitable wavelength can be chosen to pick the PPG signals of best quality. This can effectively improve the accuracy of heart rate sensing by 15%. In the literature [20,21], experimental results show that under low-temperature conditions the blood perfusion decreases, resulting in the decrease of S/N of the PPG signal. While, if the MW-PPG sensing technology is applied, the best wavelengths can be selected for users at different skin temperatures, and hence the S/N of the PPG signal can be significantly increased by 50%. Besides, it is noted that wearing the PPG sensing devices will unavoidably cause contact pressure on the skin surface, which will cause different degrees of occlusion phenomenon for the microvascular and arterioles. In other earlier studies [22,23], the authors pointed out that PPG sensing light sources at different wavelengths can be used at different degrees of occlusion phenomenon, to achieve the best S/N of the PPG signals. By using the MW-PPG sensing devices, a better S/N of the PPG signals can be obtained. Besides, since PPG signals at different wavelengths can reflect the PPG signals measured from different depths of the body, the pulse transit time (PTT) derived from PPG signals at different wavelengths can also be obtained by using MW-PPG sensing technology. It was shown that the correlation coefficient R between the PTT and blood pressure can reach more than 0.9, demonstrating the feasibility of using MW-PPG signals for blood pressure measurement [24,25].

However, in many prior studies, as illustrated in Figure 1a, spectrometers were used as the sensing devices for sensing MW-PPG signals [21,22]. We note that not only the large size of the conventional spectrometer leads to inconvenience in the measurement setup, but the high price of the conventional spectrometer also leads this approach unable to be integrated in the daily applications. Some scholars have tried to use photodiodes (PDs) such as AFE4404 chip of Texas Instruments (TI) [26], BH1790GLC chip of ROHM Semiconductor [27], and MAX30102 chip of Maxim Integrated [28], to collect PPG signals of different wavelengths, as shown in Figure 1b. This approach can effectively reduce the size and cost of the MW-PPG sensing devices. However, for example, to acquire *N* different wavelength PPG signals through this sequential sampling architecture, not only will the sampling rate of PPG signals at each wavelength be reduced by 1/*N*, but it will also require light sources of *N* different wavelengths. We note that when *N* is large, this architecture would become difficult in implementation. For practical considerations, only *N* = 2 or 3 are implemented in general.

In this study, we developed a chip-scale MW-PPG sensor using innovative an on-chip spectrometer approach, as shown in Figure 1c. The MW-PPG sensor developed is only several square micrometers. It is compact, robust, and lightweight. To provide a large number of PPG signals at different wavelengths, only one broad spectrum light-emitting diode (LED) or a few LEDs covering broad spectrum are required. Furthermore, signal processing algorithms were developed to robustly extract PPG signals using this developed MW-PPG sensing device. Experimental results show that the S/N of the maximal-ratio combined (MRC) MW-PPG signals, namely MRC-MW-PPG signals, can be increased by up to 50%, compared to those acquired from the conventional single wavelength approach. Besides, we were able to successfully demonstrate simultaneous heart rate measurement, SpO_2_ measurement and blood pressure measurement using this MW-PPG sensor developed.

The organization of the rest of this paper is stated as follows. Section 2 introduces hardware designing of the MW-PPG sensor developed. The mathematical model of the MW-PPG sensor developed and the signal processing algorithms for obtaining robust PPG signals, SpO_2_ and blood pressure are presented in Section 3. Section 4 discusses the experimental results, and we draw a conclusion in Section 5.

## 2. Design of the Developed MW-PPG Sensor

The scheme of the chip-scale MW-PPG sensor developed is shown in Figure 2. The core technology of this sensor is based on plasmonic filters which can be integrated onto a regular photo-detector such as a complementary metal–oxide–semiconductor (CMOS) imager. By introducing nanoscale structures on metal films, plasmonic filters can provide a unique way to control polarization and wavelength of light passing through the structures. One of the significant differences of the plasmonic filters is that the transmission wavelength can be controlled only by the lateral structures on a single layer. This makes it possible to produce a device containing different filter channels in a cost-effective manner. The single layer plasmonic metal structures can be monolithically fabricated using the standard semiconductor wafer process such as nanoimprint lithography and etching processes, which enables the low manufacturing cost for the volume applications. We note that the fabrication cost of the plasmonic filters can be as low as a few dollars at volume. It is one of the most advanced processes in making on-chip spectrometers, as reported in our previous work [29,30].

In this work, we utilized the same concept, but made a chip-scale MW-PPG sensor to synchronously detect MW-PPG signals at 15 wavelengths, including: 505 nm, 510 nm, 515 nm, 520 nm, 525 nm, 620 nm, 625 nm, 630 nm, 635 nm, 640 nm, 930 nm, 935 nm, 940 nm, 945 nm, and 950 nm. The 15 wavelengths were grouped into three regions with three major center wavelengths: 515 nm, 630 nm and 940 nm. By using the red region PPG signals centered at 630 nm and the infrared region PPG signals centered at 940 nm, the R-values could be obtained for the SpO_2_ measurement. Furthermore, by using the green region PPG signals centered at 515 nm and the infrared region PPG signals centered at 940 nm, the PTT could be extracted for the blood pressure measurements.

In the following, it is shown how the raw MW-PPG signals, the PTTs of the raw MW-PPG signals, and the PTT-compensated PPG signals are extracted from the MW-PPG sensor we developed. We note that these quantities will be used in the sequel for robust PPG measurement, SpO_2_ measurement, and blood pressure measurement. Let x(λ,k) denote the spectrum reflected from tissues emitted by the designed light sources, as shown in Figure 2. Assume x(λ,k) is shining into the developed chip-scale MW-PPG sensor, where *k* is the discrete time index. Let $f_{i}(\lambda )$ be the transfer function of the *i*-th filter in the developed chip-scale MW-PPG sensor. The raw PPG signals from the *i*-th filter can be represented as
(1)$$y_{i}(k)=s_{i}(k)+n_{i}(k),i=1,...,15$$
where $s_{i}(k)=\int f_{i}(\lambda )x(\lambda ,k)d\lambda$ is the signal component, $n_{i}(k)$ is Gaussian noise component, and $y_{1}(k),...,y_{15}(k)$ are the raw PPG signals at wavelengths 505 nm, 510 nm, 515 nm, 520 nm, 525 nm, 620 nm, 625 nm, 630 nm, 635 nm, 640 nm, 930 nm, 935 nm, 940 nm, 945 nm, and 950 nm, respectively. In the design, 505 nm PPG signal $y_{1}(k)$ is used as a reference, and the *PTT* of the *i*-th PPG signal is expressed as
(2)$$PTT_{i}=\left(\frac{1}{f_{s}}\right)\underset{\tau \in Z}{\arg\max}(\mathrm{Corr}(y_{1}(k),y_{i}(k),\tau )),\;i=1,...,15$$
where $\mathrm{Corr}(y_{1}(k),y_{i}(k),\tau )=\sum y_{1}^{*}(k)y_{i}(k+\tau ),\;i=1,...,15$ is the cross-correlation function between $y_{1}(k)$ and $y_{i}(k)$, τ is the discrete index displacement, and $f_{s}$ is the sampling rate of the developed MW-PPG sensor. As reported in [31], skin is a layer structure and blood vessels are located in different layers, for example, small arteries are located in hypodermis layer which is the innermost layer of skin, arterioles are located in dermis layer and capillaries are located in the epidermis layer. When the blood pulse generated by the heart, it will arrive at small arteries, arterioles, and capillaries in order at different times. Since light with different wavelengths can penetrate into different depths of skin, MW-PPG signals at different wavelengths reflect the signals probing to different depths of blood vessels. In other words, MW-PPG signals carry the information of pulse arrival time at different depths of blood vessels. Conventionally, pulse transit time (PTT) is considered to be the time delay between the peak of PPG signals against the R peaks of electrocardiogram (ECG) signals [32]. In this work, the pulse transit time (PTT) is defined as the time shifting between MW-PPG signals at different wavelengths, also known as local PTT [33]. The PTT-compensated PPG signals are then expressed as
(3)$${\tilde{y}}_{i}(k)={\tilde{s}}_{i}(k)+{\tilde{n}}_{i}(k),i=1,...,15$$
where ${\tilde{s}}_{i}(k)=s_{i}(k-f_{s}PTT_{i})$ and ${\tilde{n}}_{i}(k)=n_{i}(k-f_{s}PTT_{i})$.

## 3. Methods for Extraction Robust PPG Signals, SpO_2_, and Blood Pressure Measurement

Firstly, the MRC algorithm for deriving robust PPG signals from the MW-PPG signals is presented. Secondly, the method of obtaining R-values from the PTT-compensated MW-PPG signals for SpO_2_ measurement is introduced. Third, the method of using $PTT_{i}$ for blood pressure measurement is explained.

### 3.1. MW-PPG Signals Combining Methods for Extracting Robust PPG Signals

Assume the MW-PPG signals from the developed MW-PPG sensor is quasi-steady, where $E[{\tilde{s}}_{i}^{2}(k)]={\tilde{s}}_{i}^{2}(k)$. Assume the noise of the *i*-th filter is Gaussian with zero-mean and variation $\sigma_{i}^{2}$. The S/N of the *i*-th filter can be defined as
(4)$$S/N_{i}=\frac{E[{\tilde{s}}_{i}^{2}(k)]}{E\left[{\tilde{n}}_{i}^{2}(k)\right]}=\frac{{\tilde{s}}_{i}^{2}(k)}{\sigma_{i}^{2}}$$

Assume the weights to the MW-PPG signals at different wavelengths are $w_{i},\;i=1,...,15$. The MRC-MW-PPG signal from the PTT-compensated MW-PPG signals can be expressed as
(5)$$\bar{y}(k)=\sum_{i=1}^{15}w_{i}{\tilde{y}}_{\lambda_{i}}(k)=\bar{s}(k)+\bar{n}(k)$$
where $\bar{s}(k)=\sum_{i=1}^{15}w_{i}{\tilde{s}}_{i}(k)$, $\bar{n}(k)=\sum_{i=1}^{15}w_{i}{\tilde{n}}_{i}(k)$. We assume that the signal power and noise power of the MRC-MW-PPG signals can be expressed respectively as
(6)$$\left\{ \begin{array}{l} E[{\bar{s}}^{2}(k)]=E{\left(\sum_{i=1}^{15}w_{i}{\tilde{s}}_{i}(k)\right)}^{2}={\left(\sum_{i=1}^{15}w_{i}{\tilde{s}}_{i}(k)\right)}^{2}\\ E[{\bar{n}}^{2}(k)]=E{\left(\sum_{i=1}^{15}w_{i}{\tilde{n}}_{i}(k)\right)}^{2}=\sum_{i=1}^{15}w_{i}^{2}\sigma_{i}^{2} \end{array}\right.$$

Therefore, the *S/N* of the MRC-MW-PPG signals $S/N_{total}$ is defined as ${\left(\sum_{i=1}^{15}w_{i}{\tilde{s}}_{i}(k)\right)}^{2}/\left(\sum_{i=1}^{15}w_{i}^{2}\sigma_{i}^{2}\right)$. According to the well-known Cauchy-Schwarz inequality and the MRC signal combination algorithm [34], it can be shown that
(7)$$S/N_{total}\leq \sum_{i=1}^{15}S/N_{i}$$

$S/N_{total}$ can be maximized at $S/N_{total}=\sum_{i=1}^{15}S/N_{i}$ with the optimal weights $w_{i}^{*}=\frac{{\tilde{s}}_{i}(k)}{\sigma_{i}^{2}}$. The flowchart shown in Figure 3 summarizes the algorithms for obtaining the robust MRC-MW-PPG signals. We note that the computational cost of the MRC algorithm is low and is linear scaling with respective to the number of selected components. In other words, the computational complexity of the MRC algorithm implemented was O(*n*), where *n* is the number of the picked wavelengths on the multi-wavelength PPG signals acquired by the developed MW-PPG sensor.

### 3.2. MW-PPG Signal Processing Methods for SpO_2_ Measurement

SpO_2_ is defined as the measurement of the amount of oxygen dissolved in blood. Light at different wavelengths can be used to probe the absorption level of Oxygen-bound Hemoglobin (HbO_2_) and Hemoglobin (Hb). It has been widely reported that the attenuations by Hb and HbO_2_ are largely different at wavelength 660 nm, and are nearly the same at 940 nm. In other words, if using the signal at 940 nm as a normalizer, the absorption level can be clearly distinguished by watching the signal at 660 nm. 660 nm and 940 nm are then widely used for SpO_2_ measurement in the research fields as well as in industries [35,36,37,38]. From Equation (3), we note that ${\tilde{y}}_{8}(k)$ and ${\tilde{y}}_{13}(k)$ are the PPG signals at 660 nm and 940 nm, respectively. According to the Beer–Lambert law, the optical density (OD) of ${\tilde{y}}_{8}(k)$ and ${\tilde{y}}_{13}(k)$ can be defined respectively as
(8)$$\left\{ \begin{array}{l} OD_{8}=\int_{0.25}^{2.5}{\tilde{Y}}_{8}(f)df/{\tilde{Y}}_{8}(0)\\ OD_{13}=\int_{0.25}^{2.5}{\tilde{Y}}_{13}(f)df/{\tilde{Y}}_{13}(0) \end{array}\right.$$
where ${\tilde{Y}}_{i}(f)=ℑ[{\tilde{y}}_{i}(k)]$ is the frequency response of the *i*-th PPG signal, and ℑ[•] is a Fourier transform. The R-values can be associated by $R=OD_{8}/OD_{13}$. SpO_2_ can be approximated by
(9)$${\mathrm{SpO}}_{2}=aR+b$$
where, *a* and *b* are regression coefficients of the linear models. The signal processing procedure of SpO_2_ measurement using the developed MW-PPG sensor is summarized in Figure 4.

### 3.3. MW-PPG Signals Processing Methods for Blood Pressure Measurement

According to the literature [33,39,40], the $PTT_{i}$ of Equation (2) can have a high correlation with diastolic blood pressure (DBP) and systolic blood pressure (SBP). The relationship of $PTT_{i}$ and blood pressure can be established by using a linear regression model.

In this work, we computed the averaged PTT by $PTT_{avg}=\frac{1}{15}\sum_{i=1}^{15}PTT_{i}$. We associated $PTT_{avg}$ with DBP as well as $PTT_{avg}$ with SBP as follows:(10)$$\left\{ \begin{array}{l} \mathrm{SBP}=a_{SBP}PTT_{avg}+b_{SBP}\\ \mathrm{DBP}=a_{DBP}PTT_{avg}+b_{DBP} \end{array}\right.$$
where $a_{SBP}$, $b_{SBP}$ and $a_{DBP}$, $b_{DBP}$ are the regression coefficients of the linear models for DBP and SBP, respectively. The signal processing procedure of SBP and DBP measurement using the developed MW-PPG sensor is summarized in Figure 5.

## 4. Experiment Results

Figure 6a shows the MW-PPG sensing device developed. Part *a* is the developed chip-scale MW-PPG sensor introduced in Section 2. Parts *b* and *d* are green LEDs with a center wavelength at 515 nm. Parts *c* and *e* are red and infrared LEDs with center wavelengths at 630 nm and 940 nm, respectively. The LEDs’ spectra measured by a Spectrometer (Ocean Optics, USB4000) are shown in Figure 6b.

In this section, we aim to verify the functionalities of the developed chip-scale MW-PPG sensor, yet aiming at extensive medical proof. For the purposes of verification, we only acquired 10 subjects, whose ages ranged from 20 to 60 and the ratio of men to women was 7:3, with males ranging from 160 to 180 centimeters in height and females ranging from 155 to 170 centimeters in height. To demonstrate the advantages of the chip-scale MW-PPG sensor developed, a SW-PPG sensor representing a conventional signal-wavelength PPG detector was used as a reference device. To compare the stability of the PPG signals, each subject was asked to use both the MW-PPG sensor developed and the SW-PPG sensor to acquire 15 second signals. Also, to conduct a correlation analysis between the SpO_2_ and the R-values extracted from the developed MW-PPG sensing device, a blood oximetry meter (TRUST, TD-8250A) [41] was used as a reference instrument. Besides, to perform the correlation analysis between SBP, DBP against the $PTT_{avg}$ extracted from the developed chip-scale MW-PPG sensor, an upper arm blood pressure monitor (Omron, HEM-7121) [42] was used as the reference instrument. It is worth mentioning that while considering the frequency of the human heart rate pulse signal is normally around 0.25–2.5 (Hz), we used Parks-McClellan algorithm to design a 64-degree band-pass filter (BPF), with a passband of 0.3–4.0 Hz, to eliminate the out of band noise [43].

The MW-PPG signal measurement via the innovative and fully-integrated, MW-PPG sensing device we developed is illustrated in Figure 7a. To collect the MW-PPG signals, as shown in Figure 7b, firstly connect the sensing device with PC via USB cable, put the index finger on the sensing devices, and then press the “start measurement” button on the developed graphical user interface (GUI), which is based on the MATLAB R2017a platform. 15 seconds of raw MW-PPG signals $y_{i}(k),\;i=1,...,15$ will be recorded. By using Equations (2) and (3), each PPG signals’ PTT $PTT_{i},\;i=1,...,15$ and each PTT-compensated PPG signal ${\tilde{y}}_{i}(k),\;i=1,...,15$ can be extracted from $y_{i}(k),\;i=1,...,15$. Besides, the MRC-MW-PPG signal $\bar{y}(k)$ can be obtained from ${\tilde{y}}_{i}(k),\;i=1,...,15$ using the presented MRC signal combining the algorithm introduced in Section 3, Part A. Besides, the R-values are calculated from the 660 nm and 940 nm PPG signal, ${\tilde{y}}_{8}(k)$ and ${\tilde{y}}_{13}(k)$, based on the algorithm presented in Section 3, Part B. Also, the $PTT_{avg}$ is calculated from $PTT_{i},\;i=1,...,15$ according to the algorithm introduced in Section 3, Part C.

Figure 8 shows the comparison of stability of the PPG signals from a subject using the MW-PPG sensing device developed against the SW-PPG sensing device. The curves in Figure 8a,b show the overlapped PPG waveforms collected by: (i) using the SW-PPG sensing device; and (ii) using the developed MW-PPG sensing device. For better clarity, to understand the variation of the PPG waveforms at different time segments, the averaged curve and the error bar, which is the standard deviation at the certain time segment, of the overlapped PPG waveforms in Figure 8a,b, were then computed and are shown in Figure 8b,d.We observe that the averaged variation of SW PPG signal was 0.142, whereas that of MRC-MW-PPG signals was only 0.077. Compared to the reference SW-PPG sensor, the MW-PPG sensor developed could effectively reduce the averaged variation by around 50%.

Furthermore, Figure 9 shows the comparison of stability of the PPG signals among the 10 subjects, where the red and blue bars are the averaged variation of PPG signals derived from the SW-PPG sensing device and that from the MW-PPG sensing device, respectively. It shows that, in general, compared to the SW-PPG sensing device, around 50% variation reduction could be obtained in using the developed MW-PPG sensing device.

Figure 10 shows the correlation between SpO_2_ and R-values extracted from the MW-PPG sensing device developed, where the x-axis represents the R-values extracted from the MW-PPG sensor and the y-axis represents the SpO_2_ measured by the reference instrument. From the preliminary experimental results, we see that the R-value against the SpO_2_ value can deliver a high correlation coefficient up to 0.93, which matches the experimental result reported in [24]. It shows the potential of SpO_2_ measurement using the MW-PPG sensing device developed.

Figure 11 shows the correlation between SBP and DBP, against $PTT_{avg}$ which is extracted from the MW-PPG sensing device developed. In Figure 11a,b, the x-axis represents the $PTT_{avg}$ and the y-axis represents SBP and DBP, respectively. In this paper, we initiate the research to build an innovative chip-scale MW-PPG sensor for synchronously sensing MW-PPG signals. To quickly assess the potential capabilities of the device developed, rather than conducting a comprehensive medical case study which involves larger-scale budgets and time, only a simple correlation analysis was conducted as for pre-screening. Correlation coefficients R = 0.79 between $PTT_{avg}$ and SBP, and correlation coefficients R = 0.78 between $PTT_{avg}$ and DBP were observed. The $PTT_{avg}$ extracted from the MW-PPG sensing device developed show a sufficient high correlation on blood pressure, which matches the experimental results reported in [25]. The $PTT_{avg}$ extracted from the MW-PPG sensing device was useable to estimate SBP and DBP via a simple linear regression model.

We note that as mentioned in Section 1, the current available PPG sensing devices on the market are not in the structure of synchronous MW-PPG sensing, and the main functionality is for heart rate detection. Our innovative, chip-scale and fully-integrated MW-PPG sensing devices not only have the potential to provide a more stable and robust PPG signals, from the benefits of the MRC signal combining algorithm, for accurate heart rate detection, but also can simultaneously provide both R-values, for SpO_2_ detection, and $PTT_{avg}$, potentially for blood pressure detection.

As shown in Table 1, It is worth mentioning that multi-channel optical sensors are emerging on the market. For example, AMS AG recently announced low-cost optical sensors (model: AS73210) to simultaneously detect up to six wavelengths [44]. The chip-scale MW-PPG sensor developed is superior in detecting up to 15 wavelengths, and is designed for blood pressure and SpO_2_ measurements. Besides, if compared to the sequential sampling architecture currently available on the market [26,27,28], the MW-PPG sensor developed is capable of synchronously sampling PPG signals of a large number wavelengths from a full-wavelength LED or few single-wavelength LEDs. If compared to conventional spectrometers, such as using Ocean Optics STS Microspectrometer (model: STS-VIS) [45], used by the early researchers for constructing primitive MW-PPG measurement platforms with synchronous sampling architecture, the MW-PPG sensor developed can provide a competitive advantage in size and cost for daily applications.

Note that the filter responses of the MW-PPG sensor developed are shown in Figure 12, peaks at 505 nm, 510 nm, 515 nm, 520 nm, 525 nm, 620 nm, 625 nm, 630 nm, 635 nm, 640 nm, 930 nm, 935 nm, 940 nm, 945 nm, and 950 nm, with full width at half maximum (FWHM) around 40~60 nm. While achieving the channel selection purpose, cross-talk among adjacent channels is unavoidable since these filters are broad and overlapped. However, the side-lobes of these filters are well suppressed, so that a full-wavelength light source or few single-wavelength LEDs can be used. 15 PPG signals corresponding to these regions of different wavelengths can then be acquired. We note that the wavelengths of the light sources picked need to cover the sensitivity region of the implemented filters. In this work, to focus on demonstrating applications of SpO_2_ measurement and blood pressure measurement, LEDs of blue green, and the IR region were implemented as for a practical and cost-efficient implementation.

## 5. Conclusions

In this work, we initiated the development of multi-wavelength photoplethysmogram (MW-PPG) sensors, in view of lacking this kind of sensor available in the research field or on the market. Three spectral regions centered at 515 nm, 630 nm and 940 nm were used to synchronously obtain 15 PPG signals corresponding to these regions of different wavelengths, by means of fabricating cost-efficient plasmonic filters. By utilizing the maximal-ratio combined (MRC) algorithm, the proposed approach showed a 50% variation reduction when compared with the single-wavelength reference sensor. Besides, both the R-values for the SpO_2_ measurement by using the red and infrared regions, and the pulse transit time (PTT) for the blood pressure measurement by using the green and infrared regions were investigated. Preliminary experimental results showed that the correlation coefficient between the R-values and the SpO_2_ could be as high as R = 0.93. The correlation coefficients between the PTT against systolic blood pressure (SBP) and diastolic blood pressure (DBP) could reach R = 0.79 and R = 0.78, respectively. The MW-PPG sensing device developed has full potential not only in conventional PPG measurement and SpO_2_ measurement, but also in emerging blood pressure measurement for wearable devices, all in a synchronous and simultaneous manner.

## Figures and Tables

**Figure 1 sensors-19-03698-f001:**
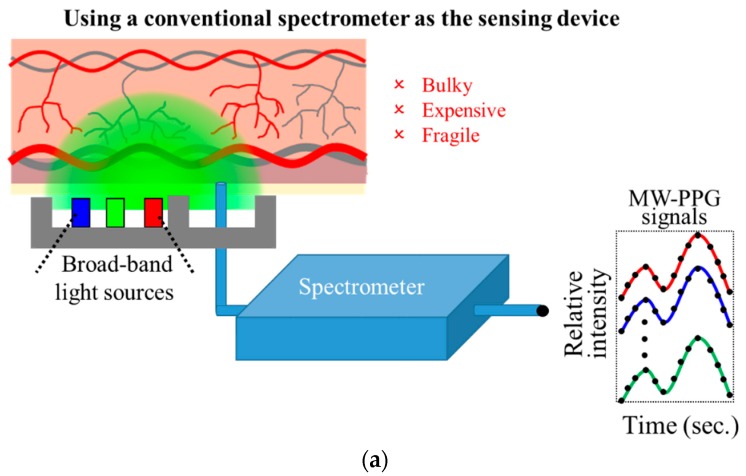
Illustration of multi-wavelength photoplethysmography (MW-PPG) sensing technologies: (**a**) prior works using a conventional spectrometer as the sensing device; (**b**) prior works using a photodiode as the sensing device in sequential sampling architecture; (**c**) the proposed work using the developed chip-scale MW-PPG sensors as the sensing device.

**Figure 2 sensors-19-03698-f002:**
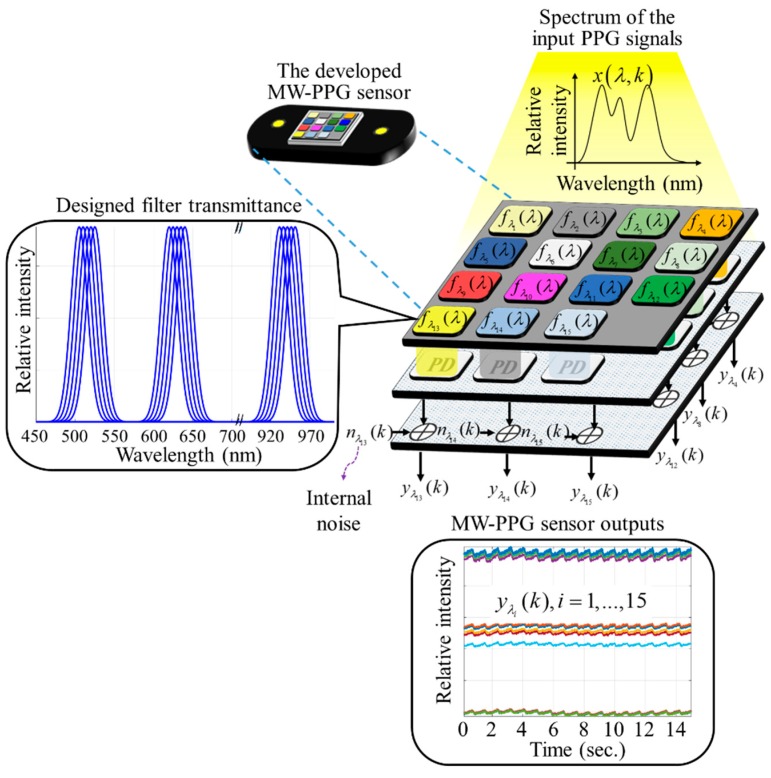
Schematic structure and mathematical model of the proposed MW-PPG sensor.

**Figure 3 sensors-19-03698-f003:**
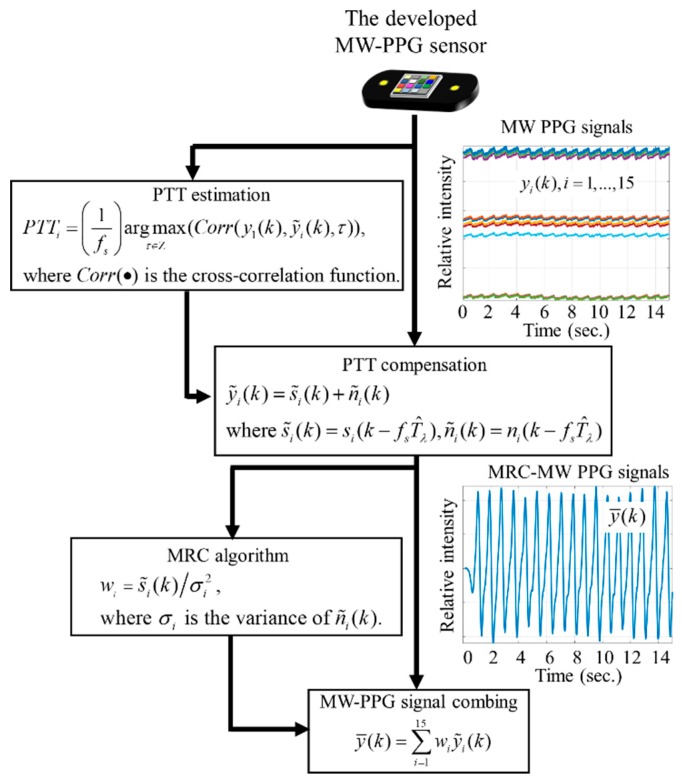
The developed signal processing flowchart for extracting robust maximal-ratio combined (MRC)-MW-PPG signals.

**Figure 4 sensors-19-03698-f004:**
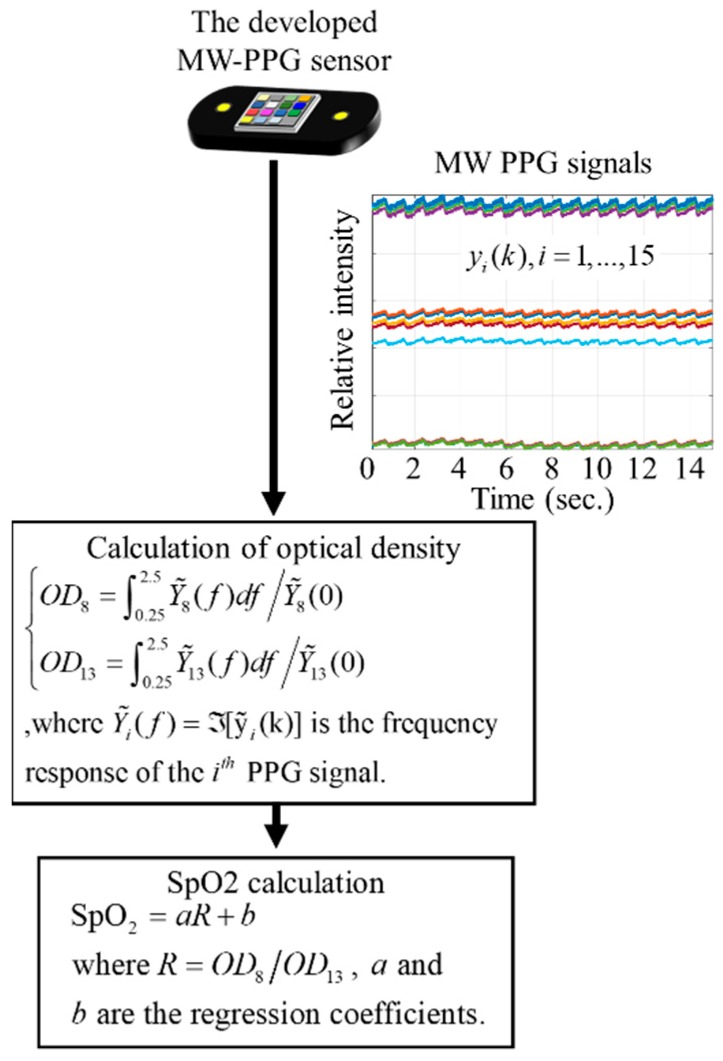
Signal processing procedure of SpO_2_ measurement using the MW-PPG sensor developed.

**Figure 5 sensors-19-03698-f005:**
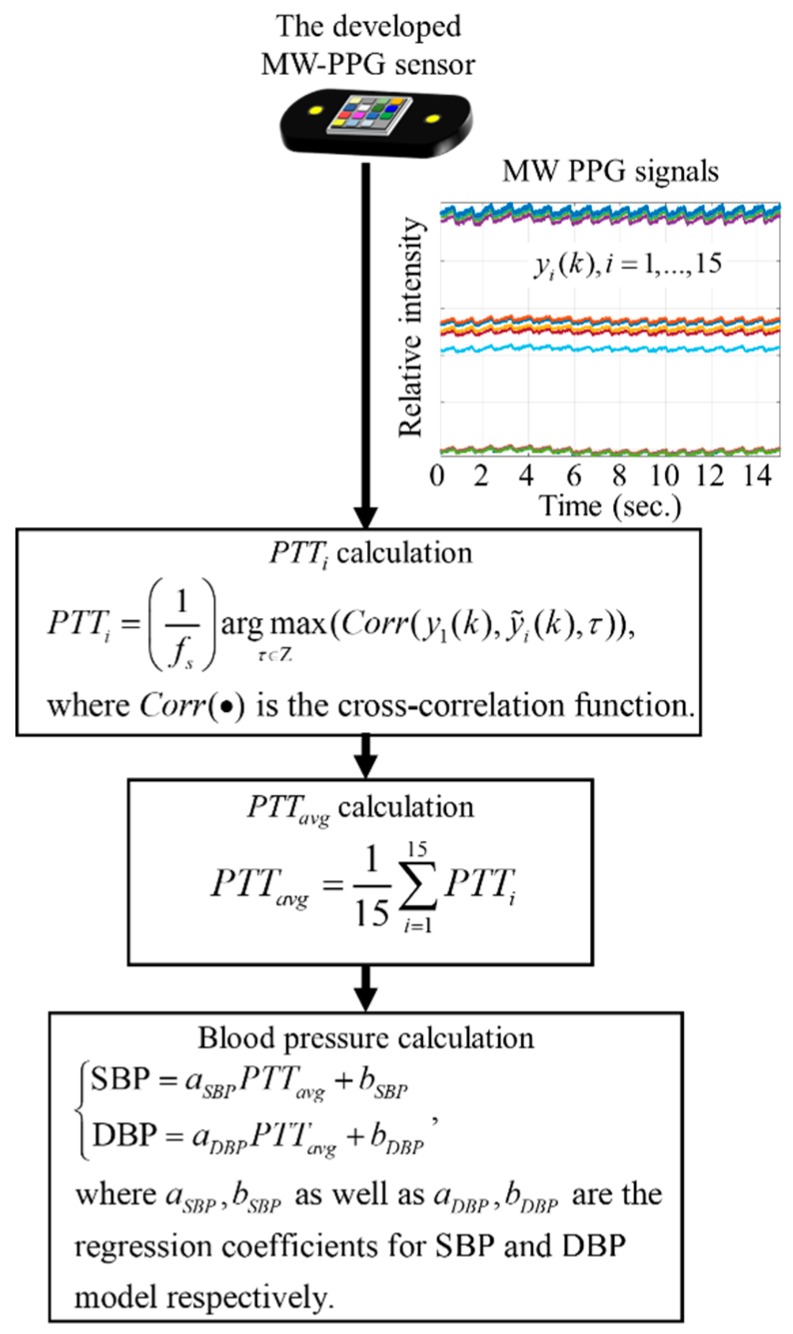
Signal processing procedure of blood pressure measurement using the MW-PPG sensor developed.

**Figure 6 sensors-19-03698-f006:**
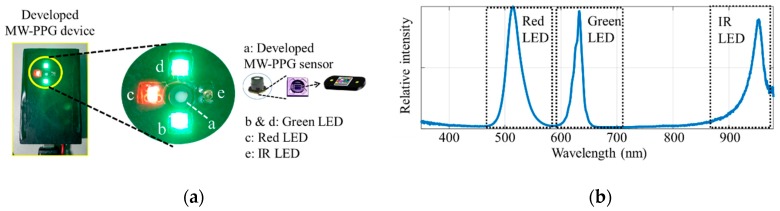
The MW-PPG sensing device we developed: (**a**) snapshot of the developed sensing device; (**b**) spectra of the light sources.

**Figure 7 sensors-19-03698-f007:**
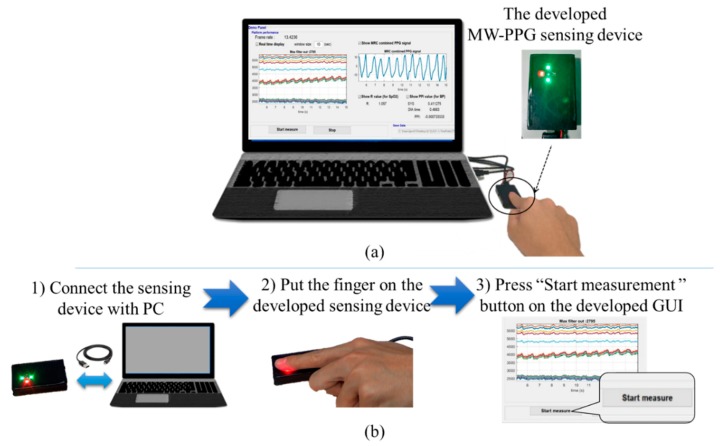
Operation of the developed MW-PPG prototype sensing device. (**a**) Snapshot of the MW-PPG measurement setup; (**b**) Illustration of the operation sequence.

**Figure 8 sensors-19-03698-f008:**
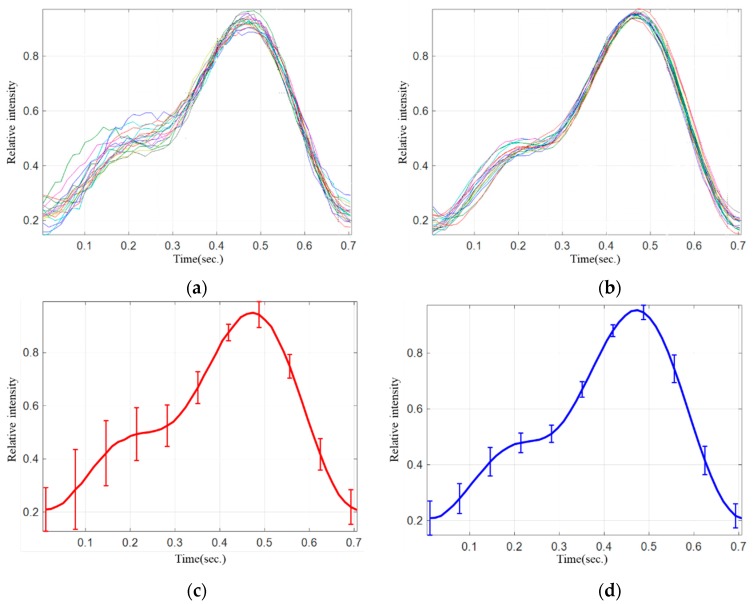
Waveform and variation of the PPG signals: (**a**,**c**) are measured by the reference SW-PPG sensing device, (**b**,**d**) measured by the MW-PPG sensing device developed.

**Figure 9 sensors-19-03698-f009:**
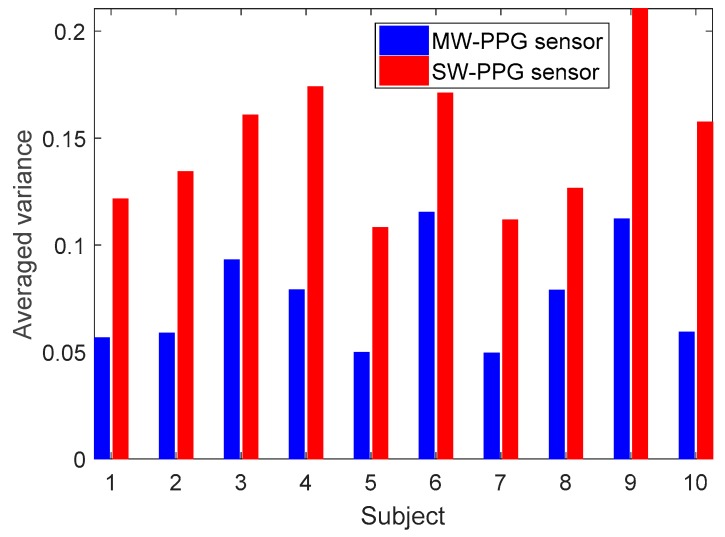
Averaged variation of 10 subjects. Red bar is derived from the reference signal-wavelength photoplethysmography (SW-PPG) sensing device, whereas blue bar is derived from the developed MW-PPG sensing device.

**Figure 10 sensors-19-03698-f010:**
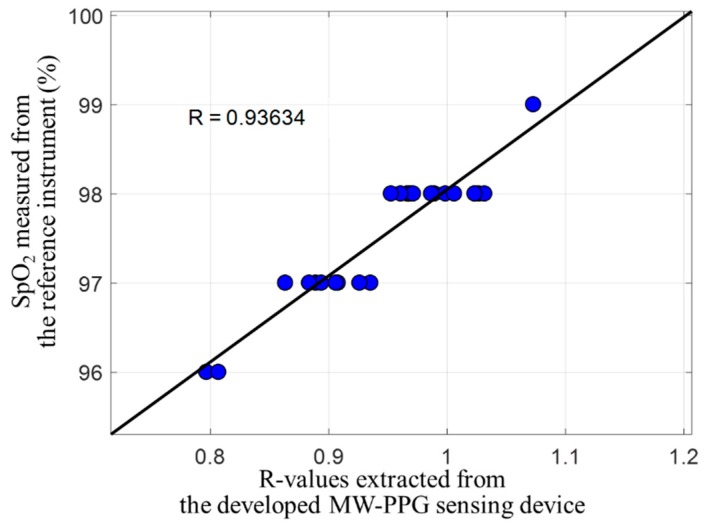
Correlation analysis between SpO_2_ and R-values extracted from the MW-PPG sensing device developed.

**Figure 11 sensors-19-03698-f011:**
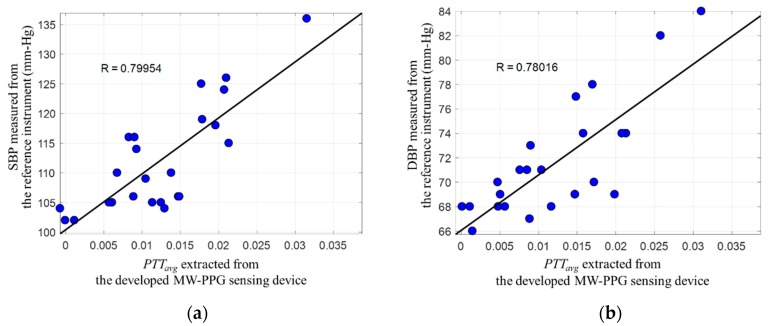
Correlation analysis between the blood pressure measured by the reference instrument against the $PTT_{avg}$ extracted from the developed MW-PPG sensing device: (**a**) systolic blood pressure (SBP); (**b**) diastolic blood pressure (DBP).

**Figure 12 sensors-19-03698-f012:**
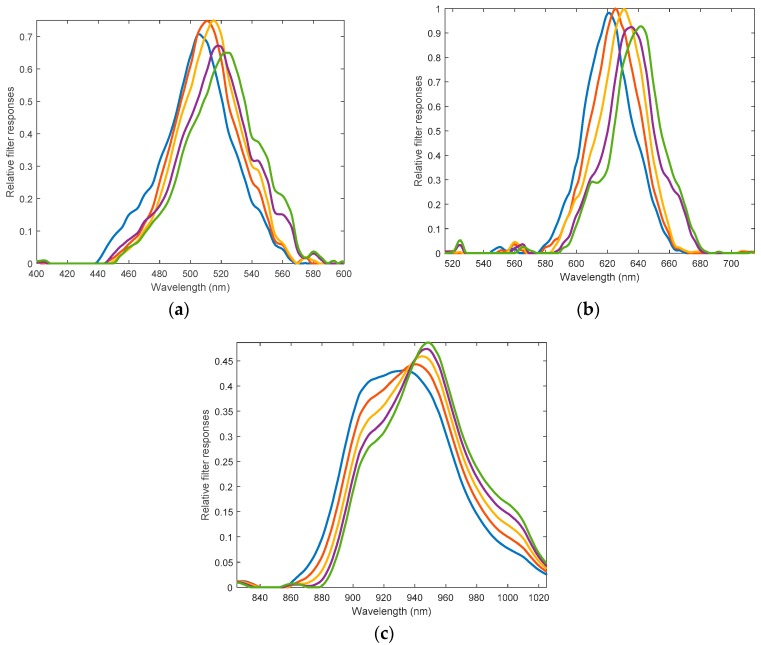
Filter responses of the 15 filters fabricated in the MW-PPG sensors developed: (**a**) in blue region; (**b**) in green region, and (**c**) in IR region.

**Table 1 sensors-19-03698-t001:** Comparison of the proposed approach against other available solutions on the market if adapted for MW-PPG sensing.

Sensor Venders	# of Channels	Synchronous/Sequential Measurement	Cost	Size
The developed MW-PPG sensor	15	Synchronous	Low	Small
AFE4404 (Texas Instruments) [26]	3	Sequential	Low	Small
BH1790GLC (ROHM Semiconductor) [27]	1	Sequential	Low	Small
MAX30102 (Maxim Integrated) [28]	2	Sequential	Low	Small
AS73210 (AMS AG) [44]	6	Synchronous	Low	Small
STS-VIS (Ocean Optics) [45]	>100	Synchronous	Very high	Very large

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
