# Peer review of "MW-PPG Sensor: An on-Chip Spectrometer Approach"

_sensors, 2019, doi:10.3390/s19173698_

Round 1
Reviewer 1 Report
The authors presented a multi-wavelength photoplethysmography (MW-PPG) sensor. Three spectral regions centered at 515 nm, 630 nm and 940 nm are used to obtain both the R-values for the SpO2 measurements (by using the red and infrared region) and the pulse transit time (PTT) for the blood pressure measurements (by using the green and infrared region). The maximal‐ratio combined (MRC) algorithm is used to achieve higher stability of PPG signals. The proposed approach shows a 50 % variation reduction when compared with the reference sensor (single-wavelength (SW-PPG) sensing device).
The paper is well written and a good set of experiments and analysis are presented. In terms of novelty, I believe the proposed approach is enough. The manuscript can be accepted for publication in Sensors MDPI Journal after the following suggestions are addressed:
Follow the MDPI template. For instance, Figure x(x) instead figure x(x). This mistake appears along the manuscript. Please, verify and correct it. The authors claim the manufacture process of plasmonic filters is cost-effective for volume applications. Could the authors give details about the fabrication price? Please, delete the additional periods (Page 4, line 101) Please, separate the values from the units. For instance, 630 nm instead 630nm. This typo appears along the manuscript. Regarding caption Figure 9. It seems the bars colors (blue and red) are inverted. Please, verify The authors claim “the developed MW‐PPG sensor is capable of synchronously sampling PPG signals of a large number wavelengths from a full‐wavelength LED or few single‐wavelength LEDs”. However, according to the prototype (2 green LED, 1 red LED and 1 infrared LED), there is no evidence of validation with full‐wavelength LED, where it is expected a cross-talk between channels deteriorating the sensor performance (since the plasmonic filters are not ideal). The author should perform some validation on that direction. What about computational cost of the implemented algorithms? In order to highlight the paper contributions, could the authors include a Table to compare the proposed approach with similar researches? Could the authors explain better the relation between the wavelengths and the variables (SpO2 and PTT)? The conclusions must be improved.
Reviewer 2 Report
The author proposed an innovative sensor setup for MW-PPG and tested its performance for SPO2 and blood pressure measurement. This study is overall well-designed, yet the authors are suggested to address the following comments.
Please check the appropriateness of the corresponding contexts for ref. [24], [25], [26] and [27]. It seems that Ref. [26] describes the MW-PPG-based blood pressure Page 9, Line 193, what is the specific passband for the filter? Page 9, Line 211, what are definitions of the variation of the PPG signal? For Fig. 8, how are the relative intensity and error bars defined? [26] and Ref. [40] are repeated. In this work, PTT was derived as the phase shift between MW-PPG signals which were collected from the same body spot. However, PTT is conventionally defined as the time delay between the proximal and peripheral pulsation signals. The author is suggested to point out the differences and rationalize the treatment in this work. The following work can serve as a reference for the authors: Liu, J., Yan, B. P., Zhang, Y. T., Ding, X. R., Su, P., & Zhao, N. (2018). Multi-wavelength photoplethysmography enabling continuous blood pressure measurement with compact wearable electronics. IEEE Transactions on Biomedical Engineering, 66(6), 1514-1525.
Round 2
Reviewer 1 Report
The authors have been addressed correctly my suggestions. The manuscript can be accepted for publication in Sensors MDPI Journal in its current form.